# Food consumption score and predictors among pregnant women attending antenatal care services in health centers of Addis Ababa, Ethiopia: Using ordinal logistic regression model

**Jerusalem Ketema Belay, Solomon Mekonnen Abebe, Lemlem Daniel Baffa, Berhanu Mengistu** [ORCID] *

Department of Human Nutrition, Institute of Public Health, University of Gondar, Gondar, Ethiopia

* berhanu5@yahoo.com

**Data Availability Statement:** all relevant data are within the manuscript and its Supporting Information files.

## Abstract

### Background

Poor maternal nutrition during pregnancy creates a stressful environment that can lead to long-term effects on tissue development. Understanding the food consumption score can be used to prevent problems associated with poor dietary intake of pregnant mothers. In Ethiopia, the food consumption score ranges from 54% to 81.5%, which is far below the World Food Program (WFP) recommendation. Thus, this study aimed to assess food consumption score and associated factors among pregnant women attending antenatal care services in health centers of Addis Ababa, Ethiopia.

### Methods

This study has used institution based cross sectional study. Overall, 999 pregnant women were selected for this study. A multistage sampling technique followed by systematic random sampling was used to include pregnant women coming for antenatal care services in the selected health centers of Addis Ababa from June 07 to July 08, 2022. We used interviewer administered questionnaire using the Kobo toolbox. Food consumption score (FCS) was assessed after collecting data on frequency of eight food groups consumed over the previous seven days, which were weighted according to their relative nutritional value. STATA 14 was used to analyse the data. Ordinal logistic regression was used to identify independent predictors of food consumption score. Those variables having p value < 0.25 in the bivariable ordinal logistic regression were considered for the final model. Crude and Adjusted Odds Ratio were used to assess the strength of the association. In the final model, p value < 0.05 at 95% confidence interval was used to declare statistical significance.

### Result

From the total of 949 pregnant women a little over half (51.20% (95%CI: 48.00%-54.40%) had acceptable food consumption score, while just over two fifth (42.60% (95% CI: 39.40%-

**Funding:** The authors received no specific funding for this work.

**Competing interests:** The authors have declared that no competing interests exist.

**Abbreviations:** FCS, Food Consumption Score; WFP, World Food Program; ANC, Antenatal Care; COR, Crude Odds Ratio; SD, Standard Deviation; AOR, Adjusted Odds Ratio; VIF, Variance Inflation Factor.

45.70%)) and a small proportion (6.2% (95%CI: 4.84%-7.94%)) of the study participants had borderline and poor food consumption score, respectively. No meal skip (AOR = 1.37, 95% CI:1.03–1.81), able to read and write (AOR = 3.99, 95% CI: 1.33–11.96), poorest wealth status (AOR = 0.52, 95% CI: 0.34–0.78), positive attitude towards consumption of a diversified diet (AOR = 1.52,95% CI: 1.17–1.98) were independent predictors of acceptable food consumption score.

## Conclusion

In this study, considerably low level of acceptable food consumption score among the study participants was observed. Besides, not skipping meal, having better educational status, wealth status and attitude towards consumption of a diversified diet were associated with acceptable food consumption score. Therefore, nutritional education considering important dietary modifications should be intensified targeting vulnerable groups.

## Background

A woman's nutritional requirements vary during pregnancy as she is now feeding both her unborn child and herself. Although prenatal nutrition has an impact on how a pregnancy develops, there is never a wrong moment to start eating healthily. Therefore, it is imperative to have a sound nutrition during the period of gestation for both the mother and her growing foetus [1–3].

However, poor maternal nutrition during pregnancy that is either due to decreased intake or quality results a range of problems [4]. It affects the general growth and development of the offspring. These changes can have a significant impact on the overall health and production performance of the offspring [5, 6]. Along with its negative impacts on the offspring's nutritional quality, it also produces a stressful environment that may have long-term or permanent repercussions on tissue development, as seen by the emergence of chronic non-communicable diseases later in life [7–9]. Understanding the food consumption score (FCS) of a pregnant woman will help to prevent the issues linked to poor dietary intake during the period of gestation [10].

Nutritional needs during pregnancy can be satisfied by eating foods from a variety of food groups including fruits, vegetables, dairy products, carbohydrates, fats, and vitamins [11]. However, poor dietary diversity and FCS have been reported during pregnancy. For example, in Bangladesh acceptable FCS among pregnant women was found to be 58%, different studies in Ethiopia have also revealed a similar figure of FCS among pregnant women:81.5% in East Gojam Zone [12], and 54% in rural Eastern Ethiopia [13], which were far below the World Food Program (WFP) recommendations (90%) [1].

A number of studies have shown the following as independent predictors of having an acceptable FCS during pregnancy: religion [12], residence [12], maternal educational status [14], educational status of the father [10], wealth status [13, 14], attitude [13], antenatal care (ANC) visit [13], skipping meal [15] and consumption of animal source food [13].

In recent years, introduction of western lifestyles in the big cities of Ethiopia like Addis Ababa has brought a drastic change in food consumption pattern of pregnant women [16], which runs counter to unrelenting efforts that is outlined in different policies and programmes enacted by the government [17, 18]. Socio-cultural factors such as women's education and employment, food preference, recent epidemics like COVID-19 and cultural practices have also been reported as driving forces for this change [19, 20]. In cognizant of this, findings from

this study can be used to provide an evidence-based decision to determine factors that influence FCS of pregnant women [21].

Even though there are a handful of researches that focused on FCS among pregnant women, our study employed a different method-ordinal logistic regression to better understand predictors of FCS among pregnant women [22]. Thus, this study aimed to assess the food consumption pattern and associated factors among pregnant women attending ANC services in health centers of Addis Ababa, Ethiopia. The goal of this study is to improve the dietary practice of pregnant women, thereby preventing long term ramifications of malnutrition.

## Methods

### Study area, design and period

The study was conducted in the capital city of Ethiopia, Addis Ababa, it is among the fastest growing cities in Africa. It was estimated that 5,228,000 people reside in the ten sub-cities of Addis Ababa in the study period [23]. The city has a sub-tropical highland climate, and is populated by people from the different regions of Ethiopia. The magnitude of food insecurity among productive safety net program beneficiaries of the city was 77.10% [24]. There were six publicly owned general hospitals and one hundred two (102) health centers, and eleven privately owned hospitals and 882 clinics in the city. By using cross-sectional study, pregnant mothers who came for ANC follow up from June 07 to July 08, 2022 at the selected health centers were approached to participate in this study. In these health centers, there were 2478 mothers who came for ANC services.

### Sample size determination and sampling technique

Sample size was estimated for each specific objective, and the highest was taken for this study. For the first specific objective, by assuming 54.46% proportion of FCS from previous study [13], 5% margin of error, 1.96 Z value at 95% confidence interval (CI) and by adding 10% non-response rate at 1.5 design effect and it was estimated to be 629. However, the highest sample size was obtained using the second specific objective. Accordingly, epi-info version 7.2.2 was used to estimate the sample size by considering the following assumptions: crude odds ratio of having acceptable FCS among pregnant women who had positive attitude towards consumption a diversified diet, which was 1.6 from a previous study [13], 80% power and 95% CI, 1.5 designs effect. Therefore, 999 was the final sample size after adding 10% non-response rate.

Pregnant women coming for antenatal care services at the selected health centers were included. However, pregnant women who were seriously ill during the data collection period were excluded in the study. Multistage sampling technique followed by systematic random sampling technique was employed to select the study participants. Out of the ten sub-cities in Addis Ababa, four sub-cities were selected randomly (30%): Nifas silk lafto sub-city, Kolfe keraniyo sub-city, Lideta sub-city and Akaki kality sub-city. In the selected sub-cities, there were 28 health centers. First, nine health centers (one from Nifas silk lafto sub-city, two from Kolfe keraniyo sub-city, three from Lideta sub-city and three Akaki kality sub-city) were selected randomly using a lottery method. Then, the required sample size was proportionated to the selected health centers, and every three (k≈ 2478/999) pregnant woman who was coming for ANC follow up was selected.

### Data collection tools and measurement

Data was collected using pretested interviewer administered questionnaire that comprises socio-demographic data, dietary habits, attitude towards consumption of a diversified diet,

obstetric history, and food consumption score (FCS). The questionnaire was first prepared in English and then translated into Amharic (Local language). We used kobo toolbox to collect the data. Nine B.Sc. nurses and four public health officers were the data collectors and supervisors, respectively. The questionnaire was pretested at 5% of the final estimated sample size at Arada sub-city. After the pre-test, the question that assessed participants' residence was excluded as all the study participants were urban residents. On the food frequency questionnaire, necessary modification was made by including foods that were not previously included.

The outcome variable of this study was food consumption score (FCS), information on foods which were consumed in the last seven days prior to the data collection time was gathered. Food consumption score (FCS) is a composite variable that is constructed based on the following criteria: food frequency, diet diversity and relative nutritional value of each food item [1]. Food consumption score (FCS) was computed after asking the study participants about the frequency and consumption of eight food groups over the period of seven days prior to the data collection period. In the questionnaire, there were 70 food items which were commonly consumed in the study area. The Cronbach's alpha value (internal consistency) was observed to be 0.82.

Then, the consumption frequencies were summed and multiplied by the standardized food group weight. Finally, it was categorized into three categories; poor food consumption score (FCS)(0–21), borderline food consumption score(FCS) (21.5–35), and acceptable food consumption score(FCS) (>35) [1, 25]. The wealth status was determined using principal component analysis which contained 15 items, and it was later categorized into five categories (Poorest to the richest) [23]. The attitude of the study participants towards the consumption diversified diet was measured using 4 item Likert-scale questions, the response ranges from strongly disagree to strongly agree. It was considered positive attitude when respondents score above the median. The internal consistency of the questionnaire was checked using Cronbach's alpha (0.78). The trimesters were defined as first trimester (less than 14 weeks), second trimester (14–27 complete weeks) and third trimester (28 complete weeks until delivery). Finally, birth interval was categorized as recommended birth interval when interpregnancy interval was more 24 months otherwise it was categorized as not recommended birth interval [26].

## Data analysis

The collected data using Kobo toolbox was exported to STATA 14 for analysis. A descriptive data was reported as frequencies, percentage, mean(±SD) and presented in tables. Ordinal logistic regression was used to identify predictors of FCS. Multicollinearity was checked using Variance inflation factor (VIF<10). Brant test of parallel regression assumption (p value = 0.66) conferred proportion of odds assumption. After checking the assumptions of ordinal logistic regression, COR and AOR at 95% was used to ascertain predictors of the outcome variable in both bivariable (p value <0.25) and multivariable ordinal logistic regression respectively. Finally, P value < 0.05 was used to determine level of significance in the final model. The final model reached after checking adequacy of the data using the Hosmer and Lemeshow test.

## Ethics approval and consent to participate

The study was conducted according to the guidelines of the 1964 Declaration of Helsinki and following amendments. Ethical clearance was obtained from University of Gondar Institutional Review Board of Institute of Public Health (Ref. No IPH/2119/2014). Permission letter was obtained from Addis Ababa Health Office. Written informed consent was obtained from

all study participants. Study participants who were unable to read and write signed by finger-prints, while doing so there were two literate witnesses. Data collectors have strictly followed COVID-19 prevention protocols. Confidentiality of the study participants was ensured; no person identifiers were used and the kobo account was password protected-only authorized user was able to access the data.

# Result

## Sociodemographic characteristics of the study participants

In this study, 949 pregnant women consented to participate in the study period, yielding 95% response rate. The vast majority of the study participants (96.80%) were married. The mean (±Standard deviation (SD)) age of the study participants in years was 27.16(±4.46SD), about two fifth (39.10%) of the study participants were in the age range 25–29 years. Regarding educational status, half of (50.10%) the study participates had accomplished primary education. More than two fifth (43.90%) of the study participants were housewives. Almost a quarter (24.50%) of the participants were from poor households. More than half (57.60%) of pregnant women have positive attitude towards consumption of variety of food (Table 1).

## Maternal characteristics

As to the maternal characteristics the study participants, more than half (57.6%) were multi-gravida, almost two third (61.5%) were in the second trimester pregnancy, more than two third (67.02%) had at least one ANC visit, and 69.6% had received nutritional counselling when they came for ANC visit (Table 2).

## Dietary habits of the study participants

Of the study participants, less than half (45.3%) ate three times a day, whereas over half (56.5%) regularly ate snacks. Nearly two thirds (59.2%) skipped meals, with the most common reasons being fatigued or preoccupied with work (19.6%), not wanting to gain weight (19.6%), and other (31.3%) causes such as loss of appetite, vomiting, and discomfort. Likewise, nearly one-third (31.1%) reported a history of food taboos. Lastly, more than a quarter (26.7%) of study participants reported having a history of food cravings (Table 3).

## Food consumption pattern

In this study, practically all of the study participants had consumed common staples, and nearly three quarters (73.2%) of the participants had consumed animal-source food, such as meat (Table 4).

## Food consumption score

This study has revealed that a little over half [51.20% (95%CI: 48.00%-54.40%)] had acceptable food consumption score. More than two fifth [42.60% (95% CI: 39.45%-45.74%)] had border-line food consumption, and the small proportion [6.2% (95%CI: 4.84%-7.94%) (Table 5).

## Factors associated with food consumption score

Ordinal logistic regression was used to identify factors associated with food consumption score. The following variables which were significant in the bivariable analysis (p value<0.25): age, husband educational status, husband occupation, maternal education, attitude, wealth status, family size, meal skip, food avoid, food craving, taking supplements, still birth, ANC visit,

**Table 1. Sociodemographic characteristics of the study participants(n = 949) in the selected health centers of Addis Ababa, Ethiopia, 2022.**

| Variables | Frequency (%) |
|---|---|
| Age (years) | |
| 18–24 | 294(31%) |
| 25–29 | 371(39.1%) |
| 30–34 | 219(23.1%) |
| ≥35 | 65(6.5%) |
| Marital status | |
| Currently married | 919(96.8%) |
| Currently unmarried | 30(3.2%) |
| Mother Educational status | |
| Unable to read and write | 30 (3.1%) |
| Read and write | 33 (3.5%) |
| Primary | 475(50.1%) |
| Secondary | 294(31%) |
| Higher (college and above) | 117(12.3%) |
| Occupation | |
| Housewife | 417(43.9%) |
| Employed(private/public) | 161(17%) |
| Merchant | 254(26.8%) |
| Daily labourer | 98(10.3%) |
| Student | 19(2%) |
| Husband educational status | |
| Unable to read to write | 24(2.5%) |
| Can read and write | 23(2.4%) |
| Primary | 298(31.4%) |
| Secondary | 385(40.6%) |
| Higher (college and above) | 219(23.1%) |
| Husband occupation | |
| Employed | 247(26%) |
| Merchant | 435(45.8%) |
| Daily labourer | 219(23.1%) |
| Driver | 48(5.1%) |
| Family size | |
| ≤4 | 82(8.6%) |
| >4 | 867(91.4%) |
| Wealth status | |
| Poorest | 197(20.8%) |
| Poor | 233(24.6%) |
| Middle | 100(10.5%) |
| Rich | 221(23.3%) |
| Richest | 198(20.8%) |
| Attitude, diversified diet | |
| Positive | 547(57.6%) |
| Negative | 402(42.4%) |

**Table 2. Maternal characteristics of the study participants(n = 949) in the selected health centers of Addis Ababa, Ethiopia, 2022.**

| Variables | Frequency (%) |
|---|---|
| Gravidity | |
| Primigravida | 402(42.2%) |
| Multigravida | 547(57.6%) |
| Birth interval | |
| Recommended | 55(10.1%) |
| Not recommended | 492(89.9%) |
| Trimester | |
| 1$^{st}$ trimester | 9(1.0%) |
| 2$^{nd}$ trimester | 584(61.5%) |
| 3$^{rd}$ trimester | 355(37.5%) |
| History of early ANC visit for the current pregnancy | |
| Had ANC | 636(67.02%) |
| No ANC | 313(32.98%) |
| Got nutrition counselling | |
| Yes | 660(69.6%) |
| No | 289(30.4%) |
| History of still-birth/abortion | |
| Yes | 59(6.3%) |
| No | 890(93.7%) |

**Table 3. Dietary habits of the study participants (n = 949) in the selected health centers of Addis Ababa, Ethiopia, 2022.**

| Variables | Frequency (%) |
|---|---|
| Meal per day | |
| Once | 3(0.3%) |
| Twice | 36(3.8%) |
| Three times | 430(45.3%) |
| Four times | 396(41.7%) |
| Five and above | 84(8.9%) |
| Habit of eating snack | |
| Yes | 536(56.5%) |
| No | 413(43.5%) |
| Meal skip | |
| Yes | 562(59.2%) |
| No | 387(40.8%) |
| Fasting while pregnant | |
| Yes | 302(31.8%) |
| No | 647(68.2%) |
| Avoid certain foods | |
| Yes | 286(30.1%) |
| No | 663(69.9%) |
| History of craving | |
| Yes | 253(26.6%) |
| No | 696(73.4%) |

**Table 4. Types of foods consumed by the study participants (n = 949) in the selected health centers of Addis Ababa, Ethiopia, 2022.**

| Type of Food | Frequency (%) |
|---|---|
| Main staples | 940(99%) |
| Pulse | 921(97%) |
| Fruits | 848(89.3%) |
| Vegetable | 906(95.4%) |
| Meat | 695(73.2%) |
| Milk | 746(78.6%) |
| Oil | 672(70.8%) |
| Sweets | 929(97.9%) |

and nutrition counselling during ANC follow-up were fitted in the final model. However, only meal skip, maternal education, attitude and wealth status were found to be the independent predictors of food consumption score.

The odds of having acceptable food consumption score among study participants who can read and write was 3.99 (Relative to borderline and poor food consumption score) times higher than study participants who were unable to read and write [AOR = 3.99,95%CI: 1.33–11.96]. The odds of having acceptable food consumption score were 49% (Relative to borderline and poor food consumption score) lower among study participants who came from the poorest households when compared to participants who came from the richest households [AOR = 0.52, 95%CI: 0.24–0.78]. The odds of having acceptable food consumption score were 1.36 times higher among study participants who did not skip meal (Versus borderline and poor food consumption score) compared to participants who skipped meal [AOR = 1.36, 95% CI: 1.03–1.81]. Finally, among study participants with positive attitude towards consumption of diversified diet there was 52% increased odds to have acceptable food consumption score (Relative to borderline and poor food consumption score) [AOR = 1.52,95%CI: 1.17–1.98] (Table 6).

**Table 5. Food consumption score of the study participants(n = 949) in the selected health centers of Addis Ababa, Ethiopia, 2022.**

| Food groups | | Food consumption score | | |
|---|---|---|---|---|
| | | Poor | Borderline | Acceptable |
| Main staples(cereals) | Yes | 57 | 402 | 481 |
| | No | 2 | 2 | 5 |
| Pulses | Yes | 57 | 388 | 476 |
| | No | 2 | 16 | 10 |
| Fruits | Yes | 28 | 346 | 474 |
| | No | 31 | 58 | 12 |
| Vegetables | Yes | 47 | 380 | 479 |
| | No | 12 | 24 | 1 |
| Meat | Yes | 8 | 236 | 451 |
| | No | 51 | 168 | 35 |
| Milk | Yes | 11 | 270 | 465 |
| | No | 48 | 134 | 21 |
| Oil | Yes | 24 | 245 | 403 |
| | No | 35 | 159 | 83 |
| Sweets | Yes | 54 | 393 | 482 |
| | No | 5 | 11 | 4 |

**Table 6. Factors associated food consumption score among pregnant women having ANC follow up at health centers of Addis Ababa (n = 949), Addis Ababa, Ethiopia, 2022.**

| Variables | Categories | Number (%) | COR (95% CI) | AOR (95%CI) |
|---|---|---|---|---|
| Age(years) | 18–24 | 294(31.00%) | 0.67(0.40–1.14) | 0.82(0.44–1.55) |
| | 25–29 | 371(39.10%) | 0.89(0.54–1.49) | 0.871(0.47–1.59) |
| | 30–34 | 219(23.10%) | 0.94(0.55–1.61) | 0.878(0.48–1.59) |
| | ≥35 | 65(6.50%) | 1.00 | |
| Husband education | Unable to read and write | 24(2.50%) | 1.00 | |
| | Read and write | 23(2.40%) | 0.64(0.46–0.89) | 0.63(0.38–1.07) |
| | Primary | 298(31.40%) | 0.62(0.44–0.87) | 0.64(0.38–1.12) |
| | Secondary | 385(40.60%) | 0.43(0.19–0.99) | 0.54(0.20–1.44) |
| | College and above | 219(23.10%) | 0.35(0.15–0.82) | 0.47(0.17–1.29) |
| Husband occupation | Daily labourer | 219(23.10%) | 1.00 | |
| | Employed | 247(26.00%) | 1.68(1.18–2.40) | 0.73(0.43–1.26) |
| | Merchant | 435(45.80%) | 1.41(1.03–1.94) | 1.23(0.87–1.75) |
| | Driver | 48(5.10%) | 1.64(0.89–3.03) | 1.64(0.85–3.18) |
| Family size | >4 | 867(91.40%) | 1.00 | |
| | ≤4 | 82(8.60%) | 0.80(0.58–1.10) | 0.83(0.56–1.23) |
| Maternal Education | Unable to read and write | 30(3.10%) | 1.00 | |
| | Read and write | 33(3.50%) | 4.38(1.53–12.55) | 3.99(1.33–11.96) * |
| | Primary | 475(50.10%) | 1.47(0.68–3.18) | 1.18(0.52–2.71) |
| | Secondary | 294(31.00%) | 2.52(1.15–5.51) | 1.84(0.78–4.32) |
| | College and above) | 117(12.30%) | 3.68(1.59–8.51) | 2.31(0.89–6.05) |
| Meal skip | Yes | 562(59.20%) | 1.00 | |
| | No | 387(40.80%) | 1.57(1.22–2.04) | 1.36(1.03–1.81) * |
| Food to avoid | Yes | 286(30.10%) | 1.00 | |
| | No | 663(69.90%) | 0.71(0.54–0.94) | 1.01(0.74–1.39) |
| Taking supplement | Yes | 649(68.40%) | 1.89(1.45–2.48) | 1.24(0.79–1.96) |
| | No | 300(31.60%) | 1.00 | |
| ANC follow-up | Yes | 636(67.02%) | 1.86(1.43–2.44) | 0.98(0.51–1.92) |
| | No | 313(32.98%) | 1.00 | |
| Craving | Yes | 253(26.60%) | 1.41(1.06–1.88) | 1.27(0.92–1.76) |
| | No | 696(73.40%) | 1.00 | |
| Nutrition counselling | Yes | 660(69.60%) | 1.93(1.47–2.54) | 1.56(0.82–2.97) |
| | No | 289(30.40%) | 1.00 | |
| Still birth | Yes | 59(6.30%) | 1.00 | |
| | No | 890(93.70%) | 1.54(0.93–2.57) | 1.36(0.79–2.33) |
| Wealth Status | Poorest | 197(20.80%) | 0.46(0.31–0.68) | 0.52(0.24–0.78) * |
| | Poor | 233(24.60%) | 1.28(0.88–1.88) | 1.19(0.80–1.76) |
| | Middle | 100(10.50%) | 0.55(0.34–0.88) | 0.69(0.43–1.14) |
| | Rich | 221(23.30%) | 0.80(0.55–1.17) | 0.85(0.17–0.58) |
| | Richest | 198(20.80%) | 1.00 | |
| Attitude | Positive | 547(57.60%) | 1.72(1.34–2.22) | 1.52(1.17–1.98) * |
| | Negative | 402(42.40%) | 1.00 | |

* Indicates p value<0.05

## Discussion

This study sought to examine FCS and associated factors among pregnant women who were having ANC follow up in health centers of Addis Ababa, Ethiopia. The results of this study have showed a little over half (51.20%, 95% CI: 48.00%-54.30%) of the study participants had acceptable FCS, and the small proportion of the study participants had poor FCS (6.20%).

Our report was far below the WFP recommendation [1]. Furthermore, the finding has showed that the percentage of acceptable FCS was comparatively lower than studies from Bangladesh(58%) [27], Nigeria (80.3%) [28] and pocket studies from Ethiopia (81.5% and 54.6% at Shegaw Motta and Eastern Ethiopia, respectively) [12, 13]. The study period could explain the decreased rate FCS, for example, the study at Shegaw Motta was conducted in the main harvest season while our study was conducted in fasting season when there is a decreased consumption animal source food [29]. Methodologically, the use of larger sample size in the current study and difference in outcome ascertainment might explain decreased rate of acceptable FCS in this study. In Ethiopia, pregnant women avoid foods due to cultural and religious reasons, and this might explain the discrepancy between the current the study and study from Nigeria where religion and culture has lesser influence over their food choice [30].

As to the associated factors of FCS, our study has showed maternal educational status-able to read and write, not being in the poorest wealth status, positive attitude towards dietary diversity, and skipping meal were independent predictors of FCS. Those mothers who were able to read and write had higher odds of having acceptable FCS compared to mothers who were unable to read and write, emphasizing the importance of nutritional educational during pregnancy. This was supported by other similar studies conducted in Nigeria [30], Ghana [31], and other studies in Ethiopia [32]. It is evident that increasing level of literacy is crucial to mitigate the problem even in the poorest households [33]. Besides, mothers who are able to read and write will have a better access to nutritional information from internet, brochures, newspapers and magazines [34–36]. In the affluents, where the toll of non-communicable disease is spiralling- enhancing level of literacy will play a pivotal role for an appropriate food selection and consumption too [37].

Being in the poorest wealth status decreases the odds of having acceptable FCS by 49% when compared mothers from the richest wealth status. This was also observed in previous studies conducted in Bishoftu, Oromia [10]. Pregnant mothers from the poorest households have limited economical accesses to procure and buy a diversified diet. On top of this, different studies have pinpointed that being in the lowest wealth status is associated with decreased consumption of animal source food [38], which in turn results lower FCS. Mothers who did not skip meal had also higher odds of having acceptable FCS when compared to their counterparts. A similar finding was observed from a study in Eastern Ethiopia [15]. During the period of gestation, meal patterning is highly important since pregnant women who sustain prolonged periods of time without food by skipping meals or snacks may be inducing a physiologic stress in their pregnancy [39]. Even though accidentally skipping a meal is not going to be harmful, skipping meals regularly for different reasons is not advisable to have a better pregnancy outcome [40, 41]. Moreover, from different studies, it has been seen that skipping meals during this period is associated decreased dietary quality [15].

The study has also revealed, study participants who had positive attitude towards consumption diversified diet had an increased odds of having acceptable FCS than their counter parts. A similar finding was observed from a study conducted in Eastern Ethiopia [13]. Different researches have supported that pregnant women with increased level of attitude have a better practice of consuming a diversified diet [42]. Women with positive feeling towards a diversified diet are also motivated to consume foods from different food groups [43, 44].

It should be mentioned that the present study has provided greater evidence on the dietary quality and predictors among pregnant women using ordinal logistic regression [22]. However, methodological limitation of the study cannot go unnoticed. Despite the use of probes like photographs-to recite memory of the study participants-problem of recall bias cannot be ignored which in turn might overestimate or underestimate the result. On top of that, cross-sectional nature of the study limits detection of causal association between the outcome and predicator variables. Even though FCS is a validated tool to asses calorie intake, the tool has not been validated to measure adequacy of macronutrients and micronutrients. The use of a 4 item Likert questionnaire is another limitation of the current study, while recommending the use of a questionnaire with sufficient numbers questions.

## Implication of the study

The findings of this study can be used to implement public health policies and programmes that strive to bring a better pregnancy outcome by promoting a balanced diet to vulnerable groups of the population. Therefore, to meet the WFP recommendation of having 90% acceptable FCS, interventions need to give a due attention to mothers with lower educational status who are from a lower socio-economic status. The implications of this study can be linked to the importance nutritional educations that target to bring a positive attitude towards consumption of a diversified diet. Moreover, findings of the study imply the importance of provision of a diversified diet in deterring the sequala of malnutrition.

## Conclusion

The findings of this study have showed that only half of the study participants had acceptable FCS which is far below the WFP recommendation. Besides, able to read and write, not skipping meal, positive attitude towards the consumption variety of foods, not being from the poorest household were significantly associated with having acceptable FCS relative to borderline and poor FCS. Therefore, it is important to give a special attention to pregnant mothers with low socioeconomic status, and mothers who skip meals in order to enhance their food variety score and improve their nutritional intake.

Future researches are encouraged to investigate nutrient adequacy among pregnant women. Finally, future studies triangulated with qualitative research that investigate behavioural factors such as food taboos and norms that influence FCS among pregnant women are also encouraged.

## Supporting information

**S1 File. FCS Plos one.**
(XLSX)

## Acknowledgments

The authors of this article are grateful for the study participants without whom this would not be possible.

## Author Contributions

**Conceptualization:** Jerusalem Ketema Belay, Solomon Mekonnen Abebe, Berhanu Mengistu.

**Data curation:** Jerusalem Ketema Belay, Berhanu Mengistu.

**Formal analysis:** Jerusalem Ketema Belay, Solomon Mekonnen Abebe, Lemlem Daniel Baffa, Berhanu Mengistu.

**Investigation:** Jerusalem Ketema Belay.

**Methodology:** Jerusalem Ketema Belay, Solomon Mekonnen Abebe, Berhanu Mengistu.

**Project administration:** Jerusalem Ketema Belay.

**Resources:** Jerusalem Ketema Belay.

**Software:** Jerusalem Ketema Belay, Solomon Mekonnen Abebe, Lemlem Daniel Baffa, Berhanu Mengistu.

**Supervision:** Jerusalem Ketema Belay.

**Writing – original draft:** Jerusalem Ketema Belay, Solomon Mekonnen Abebe, Berhanu Mengistu.

**Writing – review & editing:** Solomon Mekonnen Abebe, Lemlem Daniel Baffa, Berhanu Mengistu.

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
