## [Decision Letter · Decision Letter 0]

28 Mar 2023

PONE-D-23-02910Food consumption score and predictors among pregnant women attending antenatal care services in health centers of Addis Ababa, Ethiopia: using ordinal logistic regression modelPLOS ONE

Dear Dr. Mengistu,

Thank you for submitting your manuscript to PLOS ONE. After careful consideration, we feel that it has merit but does not fully meet PLOS ONE’s publication criteria as it currently stands. Therefore, we invite you to submit a revised version of the manuscript that addresses the points raised during the review process.

ACADEMIC EDITOR:Results: Why were the proportions of FCS out of the CI?Why did you use mean or +SD?How did you determine the variables for the wealth index to be computed for further analyses?How many questions were used to assess the attitude?Indicate which changes you require for acceptance versus which changes you recommendAddress any conflicts between the reviews so that it's clear which advice the authors should followProvide specific feedback from your evaluation of the manuscript==============================

We look forward to receiving your revised manuscript.

Kind regards,

Girma Beressa, MSc, PhD fellow

Academic Editor

PLOS ONE

Journal Requirements:

Reviewers' comments:

Reviewer's Responses to Questions

**Comments to the Author**

1. Is the manuscript technically sound, and do the data support the conclusions?

Reviewer #1: No

Reviewer #2: Partly

Reviewer #3: Yes

2. Has the statistical analysis been performed appropriately and rigorously? 

Reviewer #1: Yes

Reviewer #2: No

Reviewer #3: Yes

3. Have the authors made all data underlying the findings in their manuscript fully available?

Reviewer #1: Yes

Reviewer #2: Yes

Reviewer #3: Yes

4. Is the manuscript presented in an intelligible fashion and written in standard English?

Reviewer #1: No

Reviewer #2: No

Reviewer #3: Yes

5. Review Comments to the Author

Reviewer #1: the data do not support the conclusion and which is not inline to the result and the statistical analysis lack for some independent variable and major issue regarding the food consumption score the category used

Reviewer #2: I want to appreciate the efforts of the authors to address the nutritional problems of vulnerable population groups in Ethiopia. Next here are the comments to be addressed:

1. Abstract:

1.1. Background:

Better to indicate the magnitude of the problem

Show the gaps that your study going to fulfill

1.2. Methods:

Line 33: incomplete sentence?

Line 33-35: Jargon words, rephrase it

Line 35: The duration of the study is very short only 30 days, so how this is considered for the analysis?

Indicate how to declare statistical significance of factors and measure of association used

1.3. Conclusion:

Better to restate your finding

Line 48-49: specify the areas of nutrition education to be provided

2. Background:

Show the burden or severity of the problem logically from global to local

Indicate any program or policy intervention taken previously, any success or failures

Show any gap that was answered by this study

Indicate the contributions ( theoretical or practical) of this study

Line 83-85: tell us how ordinal logistic regression makes your study unique from the others

3. Methods:

3.1. study area:

show total number of reproductive age group women, and total number of pregnant women attending ANC

Indicate the food security status, and nutritional status of pregnant women in the study area

Include the climatic condition of the study area

3.2. Sample size:

Sample size must be calculated for each objective and then the largest one must be considered

Line 99: Why you considered COR? Is that recommended?

Indicate your inclusion criteria for your study population

The sampling techniques are not clear, better to use schematic presentation of sampling techniques

3.3. Data collection tools:

your data collection tool used for attitude is not clear? did you checked the internal consistency of the tools? How did you measured attitude?

Line 127: better to provide reference used for wealth status

Line 143: Illiterate, this is offensive word.

Line 145: Specify how did you ensured their confidentiality

4. Result:

Table 1: What was your bases to say Small and large family size

Table 2: 32.98% of your study participants were not attend ANC, but you conducted this study among pregnant women attending ANC services in your study area. so where did you get those not attending ANC?

5. Discussion:

Line 228: Better to specify the variable categories of wealth status and attitude which were statistically significant

Line 259-260: Tell us how could it be?

Reviewer #3: The manuscript has tried to evaluate food consumption score among pregnant women in.

Line 29 “………to prevent these problems…….” Which problems are you talking about rewrite the statement or correct it. Again the background session lack the gap.

Line131-132: “Multicollinearity was checked using Variance inflation factor (VIF<2)” why<2? What is your reference? This needs strong justification.

132: “After checking all the assumptions of ordinal logistic regression” what are those assumptions explain them.

Line 148: your sample size was 999; only 948 individuals participated in the study. How you missed them? What is the response rate?

Discussion session missed implication of the study

Conclusion: What will be your recommendation for future research?

6. PLOS authors have the option to publish the peer review history of their article (what does this mean?). If published, this will include your full peer review and any attached files.

Reviewer #1: No

Reviewer #2: No

Reviewer #3: No

---

## [Author Response · Author response to Decision Letter 0]

22 May 2023

Dear Editor,

I, in the behalf of other authors, would like to offer our sincere gratitude for taking the initiative in editing this manuscript. In light of your constructive comments we have tried to amend the manuscript. 

Dear Editor,

Thank you for your comments, this was occurred due to a typological error. Now we have corrected these errors. For example, on the abstract section it was previously written as (4.80-5.44), but now it has been corrected this way (48.0%-54.4%). Therefore, it has been corrected the CI including FCS. The amended CI can be found on line # 47, #48, #217, # 218, #219 and # 251. 

Dear editor,

We used mean or +SD for the continuous variables which did not have extreme values. For example, age on line #186. 

We used principal component to compute after checking all the assumptions such as sampling adequacy and Kaiser-Meyer-Olkin (KMO) measures.

To assess the attitude of mothers towards the consumption variety of diets we used 4 questions. It was 4 item Likert scale questions with responses ranging from strongly agree to strongly disagree.

The following has been written in the modified manuscript from line # 158 to #162.

‘The attitude of the study participants towards the consumption diversified diet was measured using a 4 item Likert-scale questions, the response ranges from strongly disagree to strongly agree. It was considered favourable attitude when respondents score above the median. The internal consistency of the questionnaire was checked using Cronbach’s alpha (0.78). ’

Dear Reviewers,

I would also extend my heartfelt gratitude for reviewer our manuscript. The comments are enlightening, and we have tried to resolve the raised comments and questions. 

 Responses for reviewer one:

Dear Reviewer,

We would like to offer our heartfelt gratitude for being the reviewer of our manuscript, we optimist that your comments will augment the quality our manuscript. Therefore, the following are point by point response for the raised questions and comments. 

Dear Reviewer, 

What we wanted to address there is understanding food consumption pattern by assessing the magnitude and associated factors will help in preventing pregnancy related problems that might be due decreased food consumption score. 

By assessing the food consumption score of a pregnant women one can tell the calorie adequacy of foods that a pregnant woman consumes. Besides, through this kind of study we can identify who is vulnerable, and we can also determine factors linked to low level of food consumption score. 

Dear Reviewer,

Thank you for your comments now we have incorporated the following statement on the methods section of the abstract- page 2 line #39-#42.

‘Food consumption score was assessed after collecting data on frequency eight food groups consumed over the previous seven days, which were weighted according to their relative nutritional value.’

Dear Reviewer, 

Thank you for your comments. We have made amendments based on the forwarded comment. 

For example, the word ‘vitamin’ has been edited as vitamins. The capitalized letters which were not in the beginning of the sentence have also been in small letters. 

Dear Reviewer, 

We do share your concerns, our justification behind the limited information on this topic in the light of FCS. We do know that FCS is a composite variable constructed in the following criteria: food frequency, diet diversity and relative nutritional value of each food item.

Of course, we did not deny the existence of handful of studies related to our topic. For example, in Ethiopia there were two studies one was conducted in Eastern Ethiopia while the other was conducted in Northwest Ethiopia (Reference #11 and #12). However, the two studies did not use ordinal logistic regression to ascertain associated factors. 

In the two former studies, the outcome variable was considered as binary (acceptable and unacceptable food consumption score). However, food consumption score categorized into three categories: poor, borderline and acceptable FCS. Therefore, ordinal logistic regression is more precise than binary logistic regression to identify associated factors. 

From different studies, we understand that ordinal logistic is better predictor of FCS than formerly used model, binary logistic regression. Unlike binary logistic regression, ordinal logistic considers the ordered nature of the outcome variable. 

Besides, there is a limited information on FCS among pregnant mothers residing in big cities of Ethiopia as others studies are from rural and semi urban areas.

Yes, there were pregnant mothers who were seriously ill during the data collection period. We have encountered 4 mothers who were seriously ill during the data collection period. These mothers who were seriously ill during the data collection period were excluded from the study (line # 125)

Dear reviewer,

The reason for us to use a multi-stage sampling technique is that to get the study subjects we had go through different stages.

From line # 124 to # 132 the following statement has been written:

 ‘Out of the nine sub-cities in Addis Ababa, four sub cities were selected randomly (30%) In the selected sub cities there were 28 health centres. First, nine health centres were selected randomly using a lottery method. Then, the required sample size was proportionated to the selected health centres, and every three (k≈ 2478/999) pregnant woman who was coming for ANC follow up was recruited.’

The selected health centers were: Woreda 01 HC, Alembank HC, Wereda 03 HC, Teklehaymanot HC, Beletshachew HC, Kality HC, Gelan HC, Kilinto HC and Abnet HC. These health centers are scattered over Addis Ababa. 

Finally, after selecting these health centers the study subjects were selected using systematic random sampling-then every three pregnant woman coming for ANC follow up was selected, the first was selected randomly from the three using a lottery method. There, two was randomly selected, then we selected the 2nd, the 5th, 8th and it goes on until we find the final sample. Selected study subjects were included when they came for ANC services. 

Dear Reviewer,

As we have adapted the questionnaire from other studies that included rural areas, the question residency was considered as a variable. However, during the pretest we did not find a study participant who is from a rural area. 

Dear reviewer,

After the pre-test we have added some foods in some of the food groups. For example, ‘Chiko’ and ‘Chechebsa’ were added under main staples (cereals) category; and under the vegetables category, Fossolia was included after the pertest. From the cereal section, ‘Bokolo eshet’ which was not common and inaccessible during the study period. 

Besides, to support the above explanation- we have now included the following statement from line # 141to # 143 'On the food frequency questionnaire, necessary modification was made by including foods that were not previously included.’

We have also added the following statement from line #150 to # 152

‘In the questionnaire, there were 70 food items there were commonly consumed in the study area., 

Dear reviewer, 

The questionnaire contained questions which assess the sociodemographic socio-demographic data, dietary habits, attitude obstetric history, and food consumption score of the study participants (Line #134 to 136#). For example, age, marital, family size, husband education, husband occupation, educational status, occupation and wealth status were among the variable that were included under questionnaire that used to asess sociodemographic characteristics of the participants. 

Dear Reviewer,

Yes, you are right we have mentioned it in the result section. Now in the modified manuscript we have also included a statement that explain how we measured attitude from line # 158 to # 162:

‘The attitude of the study participants towards the consumption diversified diet was measured using a 4 item Likert-scale questions, the response ranges from strongly disagree to strongly agree. It was considered favorable attitude when respondents score above the median. The internal consistency of the questionnaire was checked using Cronbach’s alpha (0.78).’

Dear Reviewer, 

We appreciate the concerns. For the best of our knowledge there were two studies which assessed FCS among pregnant women in Ethiopia. The first study was conducted in Northwest Ethiopia at Shegawa Motta Hospital (reference #11), in this study WFP classification of FCS was used which is congruent to our study. The other study which was conducted in rural Eastern Ethiopia Harerege district (reference # 12), a different definition of FCS was seen - of 0-28, 28.5-42, >42.

As it has been indicated by the WFP report, the later threshold is used when there is a higher consumption of oil and sugar on daily basis.

However, neither of the study have tried to validate the tool among pregnant women in Ethiopia.

 In cognizance of this fact, we opted to use the standardized classification that is outlined by WFP, which also used by an institution-based study in Ethiopia (reference #11). 

Moreover, we have checked the internal consistency (reliability of the questionnaire), and it was found to be 0.82. 

We have added the above statement on the modified manuscript from line # 144 to #152. 

Dear Reviewer, 

We would like to thank you for pointing out this issue. We are optimist the issues that you raised here are crucial in making the finding stronger. 

Again, we have taken the forwarded comment as one limitation of the study and the following statement have been included from # 144 to #152. 

However, the following were our assumption to use FCS in this study. 

 As you know it very well there are plenty of studies in Ethiopia those aimed to assess the dietary quality during the period of gestation. Those studies in Ethiopia including the study area t assessed diet diversity among pregnant women. These methods of dietary assessment do not give weight for different food items based on their nutritional value. However, FCS gives weight for each food items by quantifying the calorie intake. 

Moreover, FCS gives weight for each food items which indicates the importance of each food group; not mention, consideration of diet diversity and food frequency over seven day period.

Therefore, FCS is a robust measure that considers other dietary methods like diet diversity. 

To strengthen the aforementioned justification the following statement has been included from line # 144 to #152.

‘Food consumption score is a composite variable that is constructed based on the following criteria: food frequency, diet diversity and relative nutritional value of each food item’

Dear Reviewer,

We have tried to explain how we conducted wealth index from line #156 to #158. 

While conducting the wealth index we have used principal component analysis. There were 15 different variables that were considered for analysis. 

As to the reason of using wealth index over income, wealth index is an important measure of someone’s living standard. Besides, it can help us to see unequal access to health services. 

Moreover, it is difficult to exactly measure the income and expenditure of individuals to ascertain their economic status even in urban settings. 

To assess the attitude of mothers towards the consumption variety of diets we used 4 questions. It was 5 item Likert scale questionnaire responses ranging from strongly agree to strongly disagree. 

To assess the attitude of mothers towards the consumption variety of diets we used 4 questions. It was 4 item Likert scale questions with responses ranging from strongly agree to strongly disagree.

The following has been written in the modified manuscript from line # 158 to # 162.

‘The attitude of the study participants towards the consumption diversified diet was measured using a 4 item Likert-scale questions, the response ranges from strongly disagree to strongly agree. It was considered favourable attitude when respondents score above the median. The internal consistency of the questionnaire was checked using Cronbach’s alpha (0.78). ’

Dear Reviewer, 

Thank you for your constructive comments. In this section of the result unable to read and write, and can read and write have been aggregated into no formal education in the modified manuscript (Table 1) page.

However, in identifying associated factors with the outcome variable the expected count for the cells in these characteristics was >5. 

Dear Reviewer, 

We have rephrased the adjective more than half as a little over half in the modified manuscript- line # 251.

Dear Reviewer, 

We have accepted your comments, the coherence has been fixed; by first writing the report from Bangladesh to Nigeria and finally writing the reports from Ethiopia. The change can be seen from the modified manuscript line # 253 to # 256.

Dear Reviewer, 

The use of FCS provides greater evidence on dietary quality than diet diversity score. The use of FCS considers not only dietary diversity and food frequency but also the relative nutritional importance of different food groups.

Besides, unlike the use of dietary diversity score (DDS) which assess dietary over a 24 hours period; FCS provides evidence over 7-day period.

Dear Reviewer, 

Thank you for raising the importance point. FCS considers the consumption different food groups over seven day period. Even though it has limitations in assessing the adequacy of macronutrients or macronutrients, it assess the relative importance different food groups. 

Therefore, assessing different food groups over seven period will help us to see the quality of foods. 

Dear Reviewer, 

Thank you for raising this important point, as we have tried to explain at #299. we have used photograph probes of the listed foods to minimize recall bias. 

Dear Reviewer,

Thank you for your comments, now we have made some modifications. 

Dear Reviewer, 

We have accepted your comments, in the modified manuscript reference 6, 27 and 31 have been replaced with latest similar sources. 

Responses reviewer two

Dear Reviewer, I would like to offer for your insightful comments. This time we have incorporated a sentence that indicates the magnitude of the problem. It can be found at line # 31 in the clear version of the manuscript. 

Dear Reviewer, 

The gaps we intended to fill has been indicated from line # 32 to # 34, which ‘to assess food consumption score and associated factors among pregnant women attending ante natal care services in health centers of Addis Ababa, Ethiopia.’

Dear Reviewer, 

Thank you for pointing out the sentence, this time we have edited the sentence as ‘Overall, 999 pregnant women were selected for this study.’

Dear reviewer, 

The word required in line number 33 was removed, and it has been rephrased as: ‘Overall, 999 pregnant women were selected for this study.’ Besides, the word recruit has been removed and replaced with a more familiar word- include. The change can be seen from line # 36-38. 

Dear reviewer,

The time duration indicated there is not the total duration of the study, but it indicates the duration the actual data collection period. 

Dear reviewer,

 We have accepted your comment, and the following statement has been included from line # 43 to # 46.

‘Crude and Adjusted Odds Ratio were used to assess the strength of the association. In the final model, p value < 0.05 was used to declare significance.’

Dear reviewer, 

We have restated our finding saying ‘Besides, not skipping meal, having better educational status, wealth status and attitude were associated acceptable food consumption score.’

The added statement can be seen on the modified manuscript from line #56-#59

Dear reviewer,

Thank you for your comments, we have modified the manuscript based on your comments. The change can be seen on line # 58 on the modified manuscript. 

Dear Reviewer, 

We have accepted you comment and we have tried to show the burden of the problem from global to local. 

The amended statement can be found from line # 78 to # 81. 

Dear Reviewer, 

Thank you for your comments, in the modified manuscript we have tried to show the actions that has been taken, and the existing gaps by citing two references- reference #17 and #18.

Moreover, this stu

---

## [Decision Letter · Decision Letter 1]

11 Jul 2023

PONE-D-23-02910R1Food consumption score and predictors among pregnant women attending antenatal care services in health centers of Addis Ababa, Ethiopia: using ordinal logistic regression modelPLOS ONE

Dear Dr. Mengistu,

Thank you for submitting your manuscript to PLOS ONE. After careful consideration, we feel that it has merit but does not fully meet PLOS ONE’s publication criteria as it currently stands. Therefore, we invite you to submit a revised version of the manuscript that addresses the points raised during the review process.

Four questions to assess affective domain are insufficient. Ensure the entire readibility of the manuscriptThe reviewers also expressed concerns about the study's reasoning/rational of the study and methodological flaws in the manuscript. I strongly apprise you to solve these major problems before resubmitting the manuscript.==============================

We look forward to receiving your revised manuscript.

Kind regards,

Girma Beressa, MSc, PhD fellow

Academic Editor

PLOS ONE

Journal Requirements:

Reviewers' comments:

Reviewer's Responses to Questions

**Comments to the Author**

1. If the authors have adequately addressed your comments raised in a previous round of review and you feel that this manuscript is now acceptable for publication, you may indicate that here to bypass the “Comments to the Author” section, enter your conflict of interest statement in the “Confidential to Editor” section, and submit your "Accept" recommendation.

Reviewer #4: (No Response)

Reviewer #5: All comments have been addressed

2. Is the manuscript technically sound, and do the data support the conclusions?

Reviewer #4: (No Response)

Reviewer #5: Yes

3. Has the statistical analysis been performed appropriately and rigorously? 

Reviewer #4: (No Response)

Reviewer #5: Yes

4. Have the authors made all data underlying the findings in their manuscript fully available?

Reviewer #4: (No Response)

Reviewer #5: Yes

5. Is the manuscript presented in an intelligible fashion and written in standard English?

Reviewer #4: (No Response)

Reviewer #5: Yes

6. Review Comments to the Author

Reviewer #4: Methodology

You have used multistage sampling but you randomly selected why you have used 1.5 designs effect?

Number of health centers didn’t mentioned. please specify number of health centers.

Analysis

Where is your ordered outcome? To conduct ordinal logistic regression and just you said that I have conducted ordinal regression no command no assumption test, no ordinal outcomes seen.

Reviewer #5: Dear Author!, my general comments on your article were list below

Between line #29 & #30 it is better if you add the statements that shows the presence of gap on the topic

In line # 33 & #34, why you prefer the word ‘required’… ‘Recruit’? it is better if you replace by the appropriate words

In line # 33, #101 & #104, when you used ‘multistage sampling techniques’, did you decided or calculate the design effect? Why?

In line # 1 your topic of research & in line #86, food consumption patter & associated…’ please fix your title of the study. The two points are different

In line # 91, and when you were responded for the 2nd reviewer question, you said “ nine sub city, four were selected first…, then nine health centers were again selected from 28 Health Centers” by what criteria you were exclude four out of total eleven sub cities of Addis Ababa?

In line # 35 & # 95, the study period and the actual data collection periods were mismatched. In my opinion, the study period duration has greater length than the actual data collection period duration. It is better if you adjust the period in a logical manner!

In line # 33, you got 999 sampled pregnant women. In line # 40 & # 148, and when you were responded to reviewer three, Of 999 sampled population you had 51 (37 + 14) missed respondents. So, when you subtract 51 from 999 your final respondents should be 948, not 949. You were mentioned language barrier as one reason for that fourteen (14) drop out respondents. In which language did you collected your data from the respondents? Why? Did you consider a diversity of languages with in the city?

Please make the numbers of decimal similar (i.e. either one or two numbers after decimal for all narratives). As an example in line #42, #161 and #181.

In line # 113, what were your criteria when selected your data collectors and supervisors? Why?

Regarding independent predictor condition of wealth status, in line # 44 ‘ being in the poorest wealth status’ Check your analysis or reference category for this variable , in line # 269 to #270 ‘not being from the poorest household’ were independent predictor of acceptable food consumption score. Which one is correct? The former or late line?

Did you have a model or image you showed for the participants to respond the like rt scale questions? Please explain or include the model or image into the document

What is your reference to categorize attitude into favorable and unfavorable rather than positive and negative attitude?

Include in your recommendation section for the young or future researchers who need to research on the area!

Do you say something about your level of acceptable food consumption score (51.2%) when compared with national acceptable food consumption score (54%)?

7. PLOS authors have the option to publish the peer review history of their article (what does this mean?). If published, this will include your full peer review and any attached files.

Reviewer #4: **Yes: **Abera Lambebo

Reviewer #5: No

---

## [Author Response · Author response to Decision Letter 1]

29 Aug 2023

Based on the editor and the reviewers comments the manuscript has been revised, and the edited manuscript has been attached.

---

## [Decision Letter · Decision Letter 2]

6 Oct 2023

PONE-D-23-02910R2Food consumption score and predictors among pregnant women attending antenatal care services in health centers of Addis Ababa, Ethiopia: using ordinal logistic regression modelPLOS ONE

Dear Dr. Mengistu,

Thank you for submitting your manuscript to PLOS ONE. After careful consideration, we feel that it has merit but does not fully meet PLOS ONE’s publication criteria as it currently stands. Therefore, we invite you to submit a revised version of the manuscript that addresses the points raised during the review process.

ACADEMIC EDITOR:Assessment of variables, results, and recommendations are not clear. Please make the entire manuscript legible.

We look forward to receiving your revised manuscript.

Kind regards,

Girma Beressa, MSc, PhD fellow

Academic Editor

PLOS ONE

Journal Requirements:

Reviewers' comments:

Reviewer's Responses to Questions

**Comments to the Author**

1. If the authors have adequately addressed your comments raised in a previous round of review and you feel that this manuscript is now acceptable for publication, you may indicate that here to bypass the “Comments to the Author” section, enter your conflict of interest statement in the “Confidential to Editor” section, and submit your "Accept" recommendation.

Reviewer #6: (No Response)

2. Is the manuscript technically sound, and do the data support the conclusions?

Reviewer #6: Yes

3. Has the statistical analysis been performed appropriately and rigorously? 

Reviewer #6: Yes

4. Have the authors made all data underlying the findings in their manuscript fully available?

Reviewer #6: (No Response)

5. Is the manuscript presented in an intelligible fashion and written in standard English?

Reviewer #6: Yes

6. Review Comments to the Author

Reviewer #6: Comments to the Author:

Title: Food consumption score and predictors among pregnant women attending antenatal care

services in health centers of Addis Ababa, Ethiopia: using ordinal logistic regression model. The manuscript is well written, and it gives good insights on predictors of the recommended acceptable food consumption score and when these factors are given good attention it will improve inadequate dietary diversity. However, I forwarded the following minor comments to improve the manuscript.

General comment

Has language and some editorial problems. Especially the results &discussion part.

Abstract

• Background: the first sentence is too long and makes it difficult to understand the message. Therefore, make it two sentences.

• Result: “not being from the poorest wealth status (AOR=0.52, 95% CI: 0.34-0.78)”. It is good to put the result as it is rather than interpret it.

• Conclusions: line 58; what attitude is associated with acceptable food consumption

score?

Methods

Data collection tools and measurement

Line 137 “…questionnaire that comprises

socio-demographic data, dietary habits, attitude obstetric history…” What is attitude obstetrics history?

Result

Table 1: Husband's educational status: Husbands who attended a primary and above educational status can’t read and write? Please re-write as that of the mother's educational status classification.

Table 2: What is the recommended birth interval and non-recommended birth interval? It is vague.

Again you classified by trimester and then by having early ANC. What is the difference between the first trimester and “early ANC”? What is the input of these classifications for you?

Factors associated with food consumption score

The classification of maternal educational status is not as that of Table 1. Which is confusing. Either classify as 1. Can’t read and write 2. Can read and write. Or you can use the mother educational status classification on your table and regress it again.

Discussion

On line 219 “The study period

could explain…” Are sure that? Have you controlled everything??? I think the language should have to be corrected.

On line 223; “…physiological …” what is the unique physiological difference between the pregnant women of your study area and the other conducted studies?

On lines 225 -226; “…However, the FCS outcome in our finding is much higher than a study conducted in the Federal University of Agriculture, Abeokuta (FUNAAB)-19.4%.” So, what is your possible explanation for this discrepancy?

On lines 228-229; “…our study has shown maternal nutritional education…were independent predictors of FCS. Is it maternal education/maternal educational status that was an independent predictor?

Conclusion

Line 274-275; “…provide intensive nutritional education prioritizing pregnant women with lower socio-economic status and wealth status…” On this particular recommendation, how that intensive nutritional education improve low socio-economic status? / What do you recommend for?

7. PLOS authors have the option to publish the peer review history of their article (what does this mean?). If published, this will include your full peer review and any attached files.

Reviewer #6: No

---

## [Author Response · Author response to Decision Letter 2]

20 Nov 2023

We would like to appreciate the editor and the reviewer for sharing their insightful remarks and experience on the betterment of the manuscript. As usual, we have found these remarks crucial in augmenting the quality of our manuscript.

This time, we have uploaded the clear manuscript and the manuscript with track changes that was updated based on your constructive comments.

---

## [Decision Letter · Decision Letter 3]

5 Dec 2023

PONE-D-23-02910R3Food consumption score and predictors among pregnant women attending antenatal care services in health centers of Addis Ababa, Ethiopia: using ordinal logistic regression modelPLOS ONE

Dear Dr. Mengistu,

Thank you for submitting your manuscript to PLOS ONE. After careful consideration, we feel that it has merit but does not fully meet PLOS ONE’s publication criteria as it currently stands. Therefore, we invite you to submit a revised version of the manuscript that addresses the points raised during the review process.

We look forward to receiving your revised manuscript.

Kind regards,

Girma Beressa, MSc, PhD fellow

Academic Editor

PLOS ONE

Journal Requirements:

Reviewers' comments:

Reviewer's Responses to Questions

**Comments to the Author**

1. If the authors have adequately addressed your comments raised in a previous round of review and you feel that this manuscript is now acceptable for publication, you may indicate that here to bypass the “Comments to the Author” section, enter your conflict of interest statement in the “Confidential to Editor” section, and submit your "Accept" recommendation.

Reviewer #6: All comments have been addressed

2. Is the manuscript technically sound, and do the data support the conclusions?

Reviewer #6: Yes

3. Has the statistical analysis been performed appropriately and rigorously? 

Reviewer #6: Yes

4. Have the authors made all data underlying the findings in their manuscript fully available?

Reviewer #6: Yes

5. Is the manuscript presented in an intelligible fashion and written in standard English?

Reviewer #6: Yes

6. Review Comments to the Author

Reviewer #6: Comments to the Author:

Title: Food consumption score and predictors among pregnant women attending antenatal care

services in health centers of Addis Ababa, Ethiopia: using ordinal logistic regression model. The manuscript is well improved. But I have some critical recommandations and comments still on the next points.

On Table 2. ANC classification pat. It is still vegue for your readers. As I understand from your explanation It is better if you say “History of early ANC vist for the current pregnancy”.

Discussion

1. On line 223; “…physiological …” what is the

unique physiological difference between the

pregnant women of your study area and the

other conducted studies?

Even though you can state some scientific explanation ,Still you persist on “physiological difference” .. As far as I know there is no physiological difference between the Addis Ababa’s pregnant women and Nigerian or some else in the world. Because they are all human beings and we are using same standered books like Wiliams, Gabe…. Therefore please make it scientific rather than gussing,

2. On lines 225 -226; “…However, the FCS

outcome in our finding is much higher than a

study conducted in the Federal University of

Agriculture, Abeokuta (FUNAAB)-19.4%.”

So, what is your possible explanation for this

discrepancy?

Your response was:

The possible explanation for this discrepancy

is:

‘This disparity may have been attributed to

Ethiopia’s recently implemented food and

nutrition policy and strategy, which aims to

increase nutrition literacy.

Ofcourse the policy may play some role but,

The next question is are you sure that the Nigerian’s have no food and nutrition policy and strategy? They have endorsed before Ethiopia on 2001.

Read on: NATIONAL POLICY (nationalplanning.gov.ng)

So, it is good to see for another possible explanation.

Recommondation

3. Page 21 # line 333-337. Please revise the language and put your reccommndation based only on your findings!

7. PLOS authors have the option to publish the peer review history of their article (what does this mean?). If published, this will include your full peer review and any attached files.

Reviewer #6: No

---

## [Author Response · Author response to Decision Letter 3]

4 Jan 2024

The manuscript is well improved. But I have some critical recommandations and comments still on the next points. Dear Reviewer, 

As always, we would like to extend our gratitude for your insightful remarks. We have tried to answer and resolve the raised remarks based on your suggestions and comments. 

We would also like to assure you that the responses we incorporated previously, and now are based on scientific evidences-either published articles or books. 

On Table 2. ANC classification pat. It is still vegue for your readers. As I understand from your explanation It is better if you say “History of early ANC vist for the current pregnancy”. Dear Reviewer,

Thank you for the suggested remark. We have accepted your comment, and it has been modified accordingly.

On the modified manuscript we have edited Table 2 as follows (Page 12):

“History of early ANC visit for the current pregnancy”. As you have suggested it, we have replaced ‘Earlier ANC visit for the current pregnancy’ with ‘History of early ANC visit for the current pregnancy’ to make the point clearer for our readers. 

1. On line 223; “…physiological …” what is the unique physiological difference between the pregnant women of your study area and the other conducted studies? Even though you can state some scientific explanation, still you persist on “physiological difference”. As far as I know there is no physiological difference between the Addis Ababa’s pregnant women and Nigerian or some else in the world. Because they are all human beings and we are using same standard books like Wiliams, Gabe…. Therefore, please make it scientific rather than guessing, Dear Reviewer,

We have accepted your comment, this time we have made edits based on your constructive comment. Therefore, we have removed physiological difference. And it has been rephrased as follows:

‘In Ethiopia, pregnant women avoid foods due to cultural and religious reasons, and this might explain the discrepancy between the current and the study from Nigeria where religion has lesser influence over their food choice.’

https://www.ncbi.nlm.nih.gov/pmc/articles/PMC8483102/

The change can be seen from line# 269-271 in the edited manuscript. 

P.S.

 Dear reviewer, again, as you have clearly stated it, there is no physiological difference between pregnant women in Ethiopia and other parts of the world. In the previous comment, what we were trying to state was the difference in the study participates. Our study participants comprise only pregnant women while the other study was conducted from different population group-Market women- and this was what we have tried to explain previously. 

 2. On lines 225 -226; “…However, the FCS

outcome in our finding is much higher than a study conducted in the Federal University of Agriculture, Abeokuta (FUNAAB)-19.4%.”

So, what is your possible explanation for this discrepancy?

Your response was: The possible explanation for this discrepancy is: ‘This disparity may have been attributed to Ethiopia’s recently implemented food and nutrition policy and strategy, which aims to increase nutrition literacy.

Of course the policy may play some role but,

The next question is are you sure that the Nigerian’s have no food and nutrition policy and strategy? They have endorsed before Ethiopia on 2001.

Read on: NATIONAL POLICY (nationalplanning.gov.ng)

So, it is good to see for another possible explanation. Dear Reviewer,

Thank you for your insightful remark, and suggestion. We have gone through national policy on food and nutrition that was developed by the national planning committee of Nigeria in 2001. In addition to the time gap between the two policies, we have seen some differences in the objectives of the two policies: videlicet, the Ethiopian policy incorporated an objective that aimed at improving Nutrition Literacy of all Ethiopians. Therefore, this might have brought a positive change on food consumption of pregnant women residing in Ethiopia. 

Moreover, after your critical remark we have seen the details of the study conducted in Nigeria- (PDF) Street food consumption score and nutritional status of staff of federal university of agriculture, Abeokuta, Ogun state (researchgate.net). In this study, 49.7% of the study were male, and the remaining were non pregnant women staffs working at the university. Therefore, in the modified manuscript we have made edits by removing it.

This change can be seen from line # 272 to # 276 in the manuscript with track changes. 

Again, we would like to reassure you that based on you suggestion we have critically reviewed the policies enacted by both African countries. 

Objectives of the National Policy on Food and Nutrition in Nigeria:

1. To improve food security at the household and aggregate levels to guarantee that families have access to adequate (both quantity and quality) and safe food to meet nutritional requirements for a healthy and active life;

2. To enhance care-giving capacity within households with respect to child feeding and child care practices, as well as addressing the care and well-being of mothers;

3. To improve the provision of human services, such as health care, environmental sanitation, education and community development; 

4. To improve capacity within the country to address food and nutrition problems; and

5. To raise understanding of the problems of malnutrition in Nigeria at all levels of society, especially with respect to its causes and possible solutions

Objectives of Food and Nutrition Policy of Ethiopia:

1. Ensure the availability and accessibility of adequate food to all Ethiopians at all times.

2. Improve accessibility, and quality of nutrition and nutrition smart health services at all stages of the life span in an equitable manner

3. Improve consumption and utilization of a diversified and nutritious diet that ensures a citizen’s optimal heath throughout their life cycle.

4. Improve the safety and quality of food throughout the value chain.

5. Reduce food and nutrient losses along the value chain.

6. Improve food and nutrition emergency risk management, preparedness and resilience systems.

7. Improve food and nutrition literacy of all Ethiopians.

Recommendation

3. Page 21 # line 333-337. Please revise the language and put your recommendation based only on your findings Dear Reviewer, 

Thank you for your insightful remark, based on your remark we have made a language edition on the recommendation. Besides, the recommendations are now solely emanated from the findings of our study.

 The edit can be seen from line # 329 to line #399.

“Therefore, it is important to give a special attention to pregnant mothers with low socioeconomic status, and mothers who skip meals in order to enhance their food variety score and improve their nutritional intake.”

---

## [Decision Letter · Decision Letter 4]

5 Mar 2024

PONE-D-23-02910R4Food consumption score and predictors among pregnant women attending antenatal care services in health centers of Addis Ababa, Ethiopia: using ordinal logistic regression modelPLOS ONE

Dear Dr. Mengistu,

Thank you for submitting your manuscript to PLOS ONE. After careful consideration, we feel that it has merit but does not fully meet PLOS ONE’s publication criteria as it currently stands. Therefore, we invite you to submit a revised version of the manuscript that addresses the points raised during the review process.

We look forward to receiving your revised manuscript.

Kind regards,

Girma Beressa, MSc, PhD fellow

Academic Editor

PLOS ONE

Journal Requirements:

Reviewers' comments:

Reviewer's Responses to Questions

**Comments to the Author**

1. If the authors have adequately addressed your comments raised in a previous round of review and you feel that this manuscript is now acceptable for publication, you may indicate that here to bypass the “Comments to the Author” section, enter your conflict of interest statement in the “Confidential to Editor” section, and submit your "Accept" recommendation.

Reviewer #5: (No Response)

Reviewer #6: All comments have been addressed

2. Is the manuscript technically sound, and do the data support the conclusions?

Reviewer #5: Yes

Reviewer #6: Yes

3. Has the statistical analysis been performed appropriately and rigorously? 

Reviewer #5: Yes

Reviewer #6: Yes

4. Have the authors made all data underlying the findings in their manuscript fully available?

Reviewer #5: Yes

Reviewer #6: Yes

5. Is the manuscript presented in an intelligible fashion and written in standard English?

Reviewer #5: Yes

Reviewer #6: Yes

6. Review Comments to the Author

Reviewer #5: Dear Author!, your manuscript was improved starting from Abstract to Reference. However, I need clarification for some points.

1. on Table 1, you classified marital status into currently married(919), and currently unmarried(30). Currently unmarried may include single, divorced, widowed etc. When I read the husbands educational status, you were included those currently unmarried numbers as well. How it possible?

2. In line#269, better if you write the percentage of acceptable FCS in the two studies done in Ethiopia.

Reviewer #6: (No Response)

7. PLOS authors have the option to publish the peer review history of their article (what does this mean?). If published, this will include your full peer review and any attached files.

Reviewer #5: No

Reviewer #6: No

---

## [Author Response · Author response to Decision Letter 4]

9 Mar 2024

Dear Author!, your manuscript was improved starting from Abstract to Reference. However, I need clarification for some points.

Dear Reviewer, 

We are grateful for your constructive comments, we would like to assure you that your insightful remarks are enlightening not only on the betterment of the current manuscript, but on our future endeavors of publishing our research works on prestigious journals like Plos one.

1. on Table 1, you classified marital status into currently married(919), and currently unmarried(30). Currently unmarried may include single, divorced, widowed etc. When I read the husbands educational status, you were included those currently unmarried numbers as well. How it possible?

Dear Reviewer, 

As you have clearly stated it, small proportion of the study participants were unmarried (3.2%). Of those, 9 (0.9%), 7 (0.7%) and 14 (1.6 %) of these unmarried mothers were single, widows and divorced, respectively. 

From those single mothers, we have asked them about the educational status of the father of the unborn child. Here, it is noteworthy that these single mothers had partners who were not officially married, at the data collection these mothers were not living with their partners (father of the unborn child) nor they were married. 

In similar vein, divorced mothers were asked this same question, the educational status of the father. We ponder there were some incidents where married couples might be divorced even after being pregnant. This has been seen in urban areas of Ethiopia. 

At to the widows, most of the widow mothers (6) were in the third trimester, only one was in the second trimester. As we go through the notes that we collected from the reports of our data collectors and the supervisors, the study participants (pregnant mothers) who were widows have told them that the fathers of the unborn children were died at most two weeks prior to the data collection period. 

Therefore, even though, the fathers of the unborn children (single and divorced mothers) were not around, in urban areas fathers have close intimacy with mothers in supporting mothers during their pregnancy and raising children after they are born. 

2. In line#269, better if you write the percentage of acceptable FCS in the two studies done in Ethiopia

Dear Reviewer, 

Thank you for your remark, we have included the percentage of acceptable FCS in the reviewed manuscript. The change can be seen on line #269 and # 270 in the modified manuscript.

---

## [Decision Letter · Decision Letter 5]

12 Jun 2024

Food consumption score and predictors among pregnant women attending antenatal care services in health centers of Addis Ababa, Ethiopia: using ordinal logistic regression model

PONE-D-23-02910R5

Dear Authors,

We’re pleased to inform you that your manuscript has been judged scientifically suitable for publication and will be formally accepted for publication once it meets all outstanding technical requirements.

Kind regards,

Girma Beressa, MSc, PhD fellow

Academic Editor

PLOS ONE

Additional Editor Comments (optional):

The authors have satisfactorily addressed the concerns; however, questions to assess attitude are insufficient and are required to be mentioned in the limitation section before publication.

Reviewers' comments:

Reviewer's Responses to Questions

**Comments to the Author**

1. If the authors have adequately addressed your comments raised in a previous round of review and you feel that this manuscript is now acceptable for publication, you may indicate that here to bypass the “Comments to the Author” section, enter your conflict of interest statement in the “Confidential to Editor” section, and submit your "Accept" recommendation.

Reviewer #5: All comments have been addressed

2. Is the manuscript technically sound, and do the data support the conclusions?

Reviewer #5: Yes

3. Has the statistical analysis been performed appropriately and rigorously? 

Reviewer #5: Yes

4. Have the authors made all data underlying the findings in their manuscript fully available?

Reviewer #5: Yes

5. Is the manuscript presented in an intelligible fashion and written in standard English?

Reviewer #5: Yes

6. Review Comments to the Author

Reviewer #5: Dear Author!,

I took a time and read your final version of your manuscript; it was very nice. Thank you

7. PLOS authors have the option to publish the peer review history of their article (what does this mean?). If published, this will include your full peer review and any attached files.

Reviewer #5: **Yes: **Tadele Amente, a PhD candidate

---

## [Editor Report · Acceptance letter]

18 Jun 2024

PONE-D-23-02910R5 

PLOS ONE

Dear Dr. Mengistu, 

I'm pleased to inform you that your manuscript has been deemed suitable for publication in PLOS ONE. Congratulations! Your manuscript is now being handed over to our production team.

Kind regards, 

on behalf of

Dr. Girma Beressa 

Academic Editor

PLOS ONE